# WearVQA: A Visual Question Answering Benchmark for Wearables in Egocentric Authentic Real-world scenarios

Eun Chang[1*], Zhuangqun Huang[1*], Yiwei Liao[1*], Sagar Ravi Bhavsar[1*], Amogh Param[1], Tammy Stark[1], Adel Ahmadyan[1], Xiao Yang[1], Jiaqi Wang[1], Ahsan Abdullah[1], Giang Nguyen[1], Akil Iyer[1], David Hall[1], Elissa Li[2], Shane Moon[1], Nicolas Scheffer[1], Kirmani Ahmed[1], Babak Damavandi[1], Rakesh Wanga[1], Anuj Kumar[1], Rohit Patel[2], and Xin Luna Dong[1]

[1]Meta Reality Labs
[2]Meta

## Abstract

We introduce **WearVQA**, the first benchmark specifically designed to evaluate the *Visual Question Answering (VQA)* capabilities of multi-modal AI assistant on wearable devices like smart glasses. Unlike prior benchmarks that focus on high-quality, third-person imagery, WearVQA reflects the unique challenges of *egocentric* interaction—where visual inputs may be occluded, poorly lit, unzoomed, or blurry, and questions are grounded in realistic wearable use cases. The benchmark comprises 2,520 carefully curated image-question-answer triplets, spanning 7 diverse image domains including both text-centric and general scenes, 10 cognitive task types ranging from basic recognition to various forms of reasoning, and 6 common wearables-specific image quality issues. All questions are designed to be answerable using only the visual input and common senses. WearVQA is paired with a rigorous LLM-as-a-judge evaluation framework with 96% labeling accuracy. Open-source and proprietary multi-modal LLMs achieved a QA accuracy as low as 24–52% on WearVQA, with substantial drops on lower-quality images and reasoning-heavy tasks. These observations position WearVQA as a comprehensive and challenging benchmark for guiding technical advancement towards robust, real-world multi-modal wearables AI systems.

## 1 Introduction

Imagine a shopper at a liquor store picking up a bottle of Rosé and asking which dishes it pairs with. Picture a DIY enthusiast kneeling under the kitchen sink, asking how to use a basin wrench to loosen the nut behind the faucet. Envision someone at a lunch table, holding a receipt and asking how much to leave as a tip.

As wearable devices move closer to mainstream adoption, they offer new opportunities to meet users' real-world needs as illustrated above. However, the egocentric perspective introduces unique challenges. The bottle might be obscured by the brim of a hat, the space under the sink might be poorly lit, and the receipt's numbers might be too small to read clearly–*egocentric images* often suffer from lower quality than the well-lit, carefully framed shots taken by smartphones. Moreover, users tend to ask questions as if the assistant shares their viewpoint, such as *"how to fix this?"* when multiple objects are visible, or *"should I use the tool on that corner to loosen the nut?"*

---

[1*]First author with equal contribution.

[2]Benchmark URL: https://huggingface.co/datasets/tonyliao-meta/WearVQA

39th Conference on Neural Information Processing Systems (NeurIPS 2025) Track on Datasets and Benchmarks.

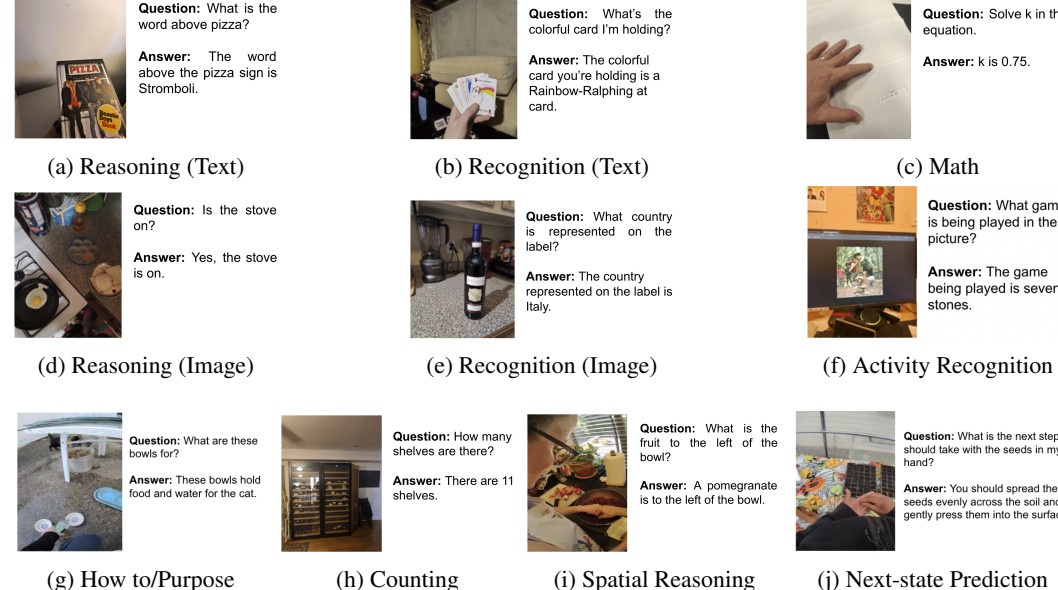

Figure 1: Example image-question-answer triples across 10 visual cognitive task types in the WearVQA benchmark.

Existing VQA benchmarks like VQA v2 [8] and OK-VQA [12] use high-quality images with clear subjects, while egocentric benchmarks such as VizWiz [9] and EgoVQA [7] capture first-person perspectives but are complexity-limited, or focus on videos. Recent benchmarks focus on specific capabilities: MMBench [10] for fine-grained abilities, MMMU [17] for expert reasoning, MMQA [14] for cross-modal information, and MathVista [11] for mathematical reasoning. This disconnect underscores the need for *a dedicated VQA benchmark based on egocentric imagery with diverse reasoning requirements, including sequential and causal reasoning* to evaluate and guide the development of state-of-the-art models in more realistic, everyday scenarios.

In this paper, we introduce **WearVQA**, which we plan to open source and build leaderboard soon. To the best of our knowledge, WearVQA is the first benchmark specifically designed for visual question answering in the context of wearable devices. WearVQA comprises 2,520 image-question-answer triples that reflect practical scenarios encountered by users of wearable devices (see examples in Figure 1). Compared to existing VQA benchmarks, WearVQA offers several distinctive features.

1. **Egocentric imagery:** All images in WearVQA are captured from a first-person per-spective, simulating the visual input typical of wearables devices. The image set includes six types of media quality issues common in such contexts—*unzoomed, occluded, rotated, cut off, blurred,* and *low light*. Notably, 54% of the images exhibit at least one of these issues (see Figure 2). In addition, 42% of the images feature *hand-holding* or *finger-pointing*, further emphasizing their egocentric nature.

2. **Egocentric questions:** Every question in the benchmark is designed to be answerable us-ing only the image and common-sense, and is egocentric to reflect the type of queries a wearables assistant might receive. The questions span ten diverse cognitive tasks, ranging from basic visual recognition (e.g., *"What am I looking at?"*), to complex tasks such as reasoning (*"Is this assembly step correct based on this manual?"*), mathematical calcula-tion (*"What is the total price after a 20% discount?"*), and spatial inference (*"What is the curvature of this road?"*), as illustrated in Figure 1.

3. **Comprehensive and insightful:** The 2,520 instances span across seven distinct image domains, ten question categories, and six image quality issues. Instances are carefully collected to achieve reasonable sizes of slices across each dimension, ensuring statistically significant metric readings.

Table 1: Comparing our benchmark to existing single-image based multimodal benchmarks in the literature.

| Benchmark | Egocentric View | Low-Quality Images | Domain Diversity | Question Diversity | Reasoning Complexity | Dataset Size (QA Pairs) | Source |
|---|---|---|---|---|---|---|---|
| VQA [6] | × | × | partial | × | Basic | 265K | COCO/abstract |
| VQA v2 [8] | × | × | partial | × | Basic | 1.4M+ | COCO dataset |
| OK-VQA [12] | × | × | partial | partial | Moderate[†] | 14,000+ | Curated COCO |
| MMBench [10] | × | × | ✓ | partial | Moderate[‡] | 2,974 | Diverse public sources |
| MMQA [14] | × | × | ✓ | ✓ | Moderate[¶] | 29,918 | Wikipedia images |
| MMMU [17] | × | × | ✓ | ✓ | Advanced[§] | 11,500 | Academic materials |
| MMVet [16] | × | × | ✓ | ✓ | Advanced[§] | 200 | Diverse web images |
| VizWiz [9] | ✓ | ✓ | ✓ | partial | Moderate | 45,000 | VizWiz mobile app |
| **WearVQA** | ✓ | ✓ | ✓ | ✓ | **Complex[*]** | **2,520** | **Consumer wearables[**]** |

[†]Knowledge integration  [‡]Attribute, relation & logic reasoning  [§]Domain expertise reasoning

[¶]Cross-modal integration  [*]**Multi-step, sequential & causal reasoning**

4. **Forward-looking:** To make sure the benchmark remains a meaningful challenge for the research community, we filtered out questions that current SOTA models consistently answer correctly. Moreover, 60% of the dataset consists of reasoning-based questions, aligned with advanced yet practical use cases expected in real-world wearable scenarios.

5. **Reliable metrics:** We carefully curate questions to be unambiguous, each with a single correct and concise answer. To enable scalable evaluation, we integrate an LLM-as-a-judge evaluation framework that scores responses based on factual correctness, relevance, completeness and conciseness. We demonstrate that this evaluation setup—using GPT-4o—achieves an accuracy of 96% in assessing answer quality.

In this paper, we present the WearVQA benchmark in detail, including the dataset construction (Section 2) and the evaluation methodology (Section 3). We use the benchmark to evaluate state-of-the-art Multi-Modal Large Language Models (MM-LLMs) (Section 4). Results show that question-answering accuracy on WearVQA ranges from 23% to 52%, and drops by 5-16% on images affected by quality issues. Our analysis further highlights key challenges, such as image quality degradations, in particular lack of zoom and occlusions, and specific question types, such as mathematical reasoning, object counting, and spatial inference, pointing to clear directions for future model improvements.

## 1.1 Related work

Recent multimodal benchmarks have advanced question-answering capabilities but fall short for wearable context. Table 1 compares our benchmark with existing multimodal benchmarks. The original VQA dataset [6] introduced the task of answering questions about images but used relatively simple queries on clean images. Traditional VQA benchmarks such as VQA v2 [8] have advanced multimodal reasoning but use high-quality images with clear subjects. Knowledge-based VQA benchmarks like OK-VQA [12] incorporate external knowledge but still rely on curated imagery. Recent comprehensive benchmarks such as MMMU [17] and MMQA [14] feature diverse domains and questions but utilize well-framed imagery from curated sources. MMVet [16] evaluates model versatility across diverse tasks with varying visual complexities but includes only partially degraded images and lacks the egocentric perspective crucial for wearable applications. MMBench [10] offers fine-grained evaluation across multiple ability dimensions but lacks complex reasoning types.

While existing benchmarks excel in specific dimensions, WearVQA uniquely combines visual challenges with multi-step reasoning requirements. These features make our benchmark a robust testbed for evaluating how multimodal systems handle real-world challenges close to typical deployment scenarios in wearable computing applications.

## 2 Data Collection

### 2.1 Problem definition

The **WearVQA benchmark** is designed to evaluate AI capabilities in understanding what are in the viewpoint of the user and answering the user's questions. Formally, the *WearVQA (Wearables*

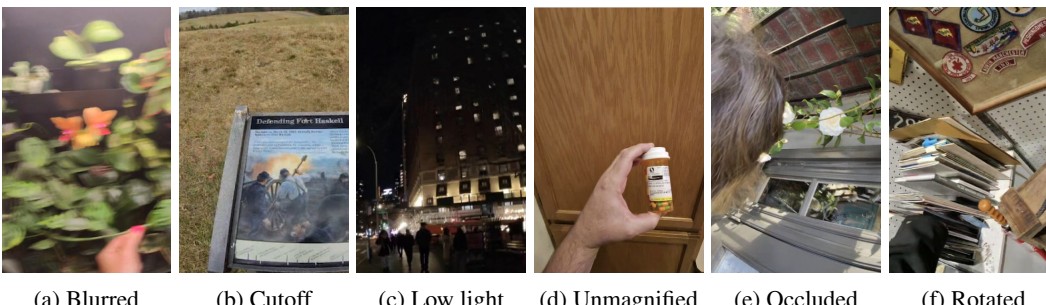

| (a) Blurred | (b) Cutoff | (c) Low light | (d) Unmagnified | (e) Occluded | (f) Rotated |

Figure 2: WearVQA contains egocentric images, where 54% images contain at least one of six quality issues common for wearables scenarios.

*Visual Question-Answering)* problem takes an *egocentric image* and an *egocentric visual question* as input, and outputs an answer based on the image. The WearVQA benchmark contains a set of image-question-answer triples, where the answer provides the ground truth for the question.

**Egocentric images:** An egocentric image is captured by a *wearable device* such as smart glasses and AI pins, from first-person perspective, simulating the viewpoint of the wearer. Our benchmark contains 7 image domains popular for wearables use cases, including *text/documents, food/drinks, landmarks/travel, shopping/products, gardening/plants, animals/pets,* and *hobbies/activities*. Egocentric images are characterized by several core defining aspects: (1) images often have wide angles and small main entities, (2) lower resolution, and (3) poor image quality, as listed below and illustrated in Figure 2.

- **Blurred:** Images that exhibit a lack of clarity and sharpness, leading to indistinct and unfocused visual elements. Blurred images are prevalent in egocentric scenarios due to the natural movements, and activities like walking, running, or gesturing which lead to blurred images.

- **Cut-off:** Images that exhibit incomplete framing, with portions of the subject matter truncated. In egocentric scenarios this occurs because the perspective is tied to the body, leading to unintentional framing.

- **Low light:** Images captured under insufficient lighting conditions, leading to diminished visibility and color accuracy. Egocentric wearables are often used in diverse environments, including low-light settings, which can adversely affect image quality.

- **Unmagnified:** Images that suffer from a lack of sufficient magnification or zoom, resulting in inadequate detail resolution. Egocentric images often have fixed focal lengths and wide-angles that capture a broad field of view, resulting in images that lack detail when subjects are at a distance.

- **Occluded:** Images where the primary object is partially obscured by intervening objects or body parts, such as hands or hair, due to the natural interferences common in egocentric captures.

- **Rotated:** Images presented with an incorrect orientation, disrupting the natural alignment. This misalignment is a result of orientation changing with the wearer's head or body movements, especially during dynamic activities.

**Egocentric visual questions:** An egocentric visual question takes a first-person perspective and asks questions regarding what is in the image. For example, a question can be *"how to use the tool on my left?"* or *"what type of flower am I holding?"*.

To make the benchmark focused, we include questions that satisfy the following four criteria. 1) *Image-based:* the question has to be answered with the image; for example, *"who wrote this book?"* instead of *"who wrote Harry Potter?"*; 2) *No external-source needed:* the question can be answered with the image and common sense, thus does not need external knowledge; for example, *"what is the price on the label?"* instead of *"is the product cheaper on Amazon?"*; 3) *Short-form:* the question can be answered with concise responses; for example, *"what is this stuff animal?"* instead of *"write*

*a poem about this stuff animal"*; 4) *Unambiguous:* the question has a single correct answer; for example, *"where is the dog relative to TV?"* instead of *"where is the dog?"*

We consider 10 types of questions (considering text and images separately) to test a diverse set of understanding and reasoning capabilities. The question types are designed to be representative of wearables use cases, with a good coverage of deep reasoning tasks. We show an example of each type in Figure 1.

- **(Text/Image) Recognition:** Questions that require identifying and recognizing texts or objects.
- **(Text/Image) Reasoning:** Questions that require logical reasoning or deduction based on commonsense knowledge, beyond simple recognition.
- **(Text) Math:** Questions that require math calculations or numerical reasoning.
- **(Image) Activity recognition:** Questions that require identifying or understanding the actions or activities being performed by individuals or groups in a given context.
- **(Image) How-to/purpose:** Questions that ask for explanations on how to perform a task, or the purpose of a tool or an action.
- **(Image) Counting:** Questions that require counting the number of objects in an image.
- **(Image) Spatial reasoning:** Questions that involve understanding and interpreting spatial relationships between objects or within an environment.
- **(Image) Next-state prediction:** Questions that involve forecasting the subsequent state of an object or scenario based on current information.

## 2.2 Data collection

**Image collection:** We captured egocentric images using RayBan Meta smart glasses[1]. We curated the image set in a structured manner, involving two key steps. (1) We instructed a group of annotators to capture images and videos from daily life, focusing on scenarios plausible for interaction with wearable devices. We also instructed annotators to include images with quality issues, such as poor lighting and occlusion, to simulate real-world conditions. (2) We sampled frames from the captured videos, such that we enrich the image set with a diverse set of visual inputs like motion blurriness. These two steps resulted in a collection of approximately 4,000 images.

**Question-answer collection:** For each question type, we decided the number of questions needed to ensure statistical significance. We instructed our annotators to write egocentric questions that are image-based, no external-source needed, short-form, and unambiguous. For each question, we instructed our annotators to write the ground truth answers in short sentences.

Finally, we removed questions where all main models discussed in Section 4 provided correct answers, to ensure a challenging benchmark. In this way we downsized to 2,520 question-answer pairs. We randomly split them into a public test set of 1,500 questions and private test set of 1,000 questions.

## 2.3 Data statistics

We created a benchmark of 2,520 image-question-answer instances, randomly split into 1,500 for a public test set and 1,000 for a private test set. The instances covers 7 domains, a total of 10 different question types, and 6 types of image quality issues. Figure 3 shows the distributions of questions along the different dimensions, and Table 2 gives details regarding types and domains. The images are of different resolutions, with ∼2.1K of resolution around 720 x 1280, and ∼413 of around 3024 x 4032, where the former represent typical size of images that can be fairly easily transferred to the server through wifi.

We have a few highlights. First, 54% of images have at least one type of quality issues, consistent with our observations in real production scenarios. Second, 48% of images have hand-holding or finger-pointing, highlighting uniqueness of egocentric images. Third, 60% of questions require

---

[1]https://www.meta.com/ai-glasses/

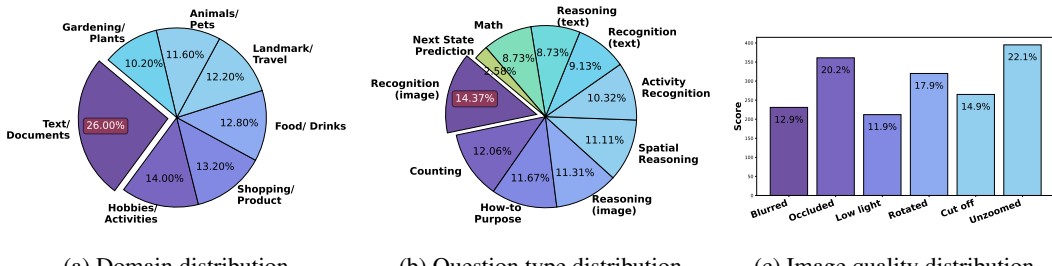

| (a) Domain distribution | (b) Question type distribution | (c) Image quality distribution |

Figure 3: Instances are mostly uniformly distributed across different domains, question types, and image quality issues in WearVQA.

Table 2: Distribution of question types across visual domains in WearVQA.

| Type | Image | | | | | | Text | Total |
|---|---|---|---|---|---|---|---|---|
| | Animals/ Pets | Food/ Drinks | Gardening/ Plants | Hobbies/ Activities | Landmark/ Travel | Shopping/ Product | Text/ Documents | |
| Recognition | 71 | 81 | 53 | 39 | 64 | 54 | 230 | **592** |
| Activity Recognition | 49 | 33 | 24 | 124 | 20 | 10 | n/a | **260** |
| How-To Purpose | 32 | 44 | 31 | 62 | 33 | 92 | n/a | **294** |
| Object Counting | 37 | 58 | 42 | 34 | 63 | 70 | n/a | **304** |
| Spatial Reasoning | 44 | 49 | 55 | 28 | 62 | 42 | n/a | **280** |
| Reasoning | 53 | 48 | 44 | 39 | 46 | 55 | 220 | **505** |
| Next-state Prediction | 4 | 7 | 6 | 24 | 17 | 7 | n/a | **65** |
| Math | n/a | n/a | n/a | n/a | n/a | n/a | 220 | **220** |
| **Domain Total** | **290** | **320** | **255** | **350** | **305** | **330** | **670** | **2520** |

various kinds of reasoning, such as counting, spatial reasoning, general reasoning, and inferential reasoning. Finally, each slice contains at least 220 instances thus guaranteeing a margin of error below 5.7% with 90% confidence, with the only exception of Next-state Prediction, which is unusual in real scenarios.

## 3 Evaluation of models

We measure the *accuracy* of VQA systems, which calculates the percentage of answers that are *correct*. Detailed prompt can be found in Appendix A.2. We illustrated using the first example question "What is the word above pizza?" in Figure 1. A correct response needs to meet all of the following five criteria.

- *Factual correctness:* The response must be factually accurate and free from hallucinations. *"The word is marinara"* gives a wrong answer.

- *Relevance:* The response must be relevant to the question. *"The book is red"* is irrelevant.

- *Completeness:* The response should directly address and fully answer the question. *"Sorry, I can't help with that."* is incomplete.

- *Egocentric:* The response should be phrased in an egocentric way. For example, *""The image shows the word Strombolli"* is not egocentric.

- *Conciseness:* The response should be brief and avoid unnecessary information; for example, *"Hello, the word is Strombolli, written in small black text in Times New Roman font."*

To investigate the performance of the LLM judge, we generated human annotations on 10% of the model responses. The LLM Judge demonstrates an accuracy of 96% in deciding answer correctness. In terms of identifying incorrect answers, its precision (percentage of real errors out of identified errors) is 98.2%. recall (percentage of identified errors out of real errors) is 95.5%, resulting with a F1-score of 96.8%. We thus can trust the LLM judge in benchmarking.

A manual audit revealed that approximately 1.2% of the dataset entries contained minor inaccuracies, primarily in attribute labeling.

Table 3: Model QA Accuracy (in percentage) on WearVQA.

| License Type | Model | QA Accuracy | High Quality Images | Low Quality Images |
|---|---|---|---|---|
| Open Source | Phi-4-mini-instruct (3.8B) | 23.9 | 26.9 | 21.3 (-5.6) |
| | Molmo-72B-0924 | 25.7 | 29.7 | 22.2 (-7.5) |
| | Pixtral-12B-2409 | 25.8 | 29.5 | 22.7 (-6.8) |
| | Llama-3.2v-90B-Vision | 37.7 | 41.6 | 33.8 (-7.8) |
| | Llama-4-Scout-17B-16E-Instruct | 41.5 | 45.6 | 38.0 (-7.6) |
| | Llama-4-Maverick-17B-128E-Instruct | 42.4 | 45.9 | 39.3 (-6.6) |
| | Qwen2.5-VL-72B-Instruct | *45.1* | 51.1 | *39.9* (-11.2) |
| Proprietary | Claude-3.7-sonnet (175B) | 33.9 | 42.6 | 26.4 (-16.2) |
| | Gemini-1.5-Pro (200B) | ***45.4*** | ***50.4*** | ***41.0*** (-9.4) |
| | GPT-4o (200B) | **51.5** | **58.5** | **45.5** (-13.0) |

# 4 Benchmarking

In this section, we conduct a systematic evaluation of state-of-the-art MM-LLMs on the WearVQA Benchmark to assess the capabilities and limitations of each model in handling the unique challenges posed by the wearables contexts. We answer two questions in this evaluation.

- **Q1:** Does the WearVQA benchmark have the right level of difficulty?
- **Q2:** Where do SOTA MM-LLMs fall short in answering wearables VQA questions?

## 4.1 Experiment setup

We evaluated a diverse set of MM-LLM models, across proprietary and open-source models, on the public test set.

- **Proprietary Models:** GPT-4o (175B), Gemini-1.5-Pro (200B), Claude-3.7-sonnet (175B)
- **Open-Source Models:** Qwen2.5-VL-72B-Instruct [15], Llama-4-Maverick-17B-128E-Instruct [2], Llama-4-Scout-17B-16E-Instruct [3], Llama-3.2v-90B-Vision [1], Phi-4-mini-instruct (3.8B) [13], Molmo-72B-0924 [4], Pixtral-12B-2409 [5]

Experiments were conducted on NVIDIA H100-SXM5-80GB GPUs with varying configurations depending on model size and computational requirements. We evaluated smaller models (Phi-4 and Pixtral) on a single H100 GPU, and mid-sized models (Qwen2.5 and Molmo) on 4 H100 GPUs. We evaluated Llama-4-Scout and Llama-4-Maverick on 8 and 16 H100 GPUs respectively. For proprietary models (GPT-4o, Gemini, Claude-3.7), we conducted evaluations through their respective API services.

## 4.2 Overall model performance

We observe that the quality of the models ranges from 23% to 52%, showing that the WearVQA benchmark exposes challenges in addressing wearables VQA needs (**Q1**). Among all models, GPT-4o stands out as the top performer with an accuracy of 52% [2], followed by Gemini-1.5-Pro with an accuracy of 45%. Among open-sourced models, Qwen-2.5-VL is ranked first with an accuracy of 45%, followed by Llama-4-Maverick with an accuracy of 42%.

On images with quality issues, in general we observe similar rankings. Phi-4, Pixtral-12B, Llama-4-Maverick are among the models with the smallest quality drop (2.5-4%), showing the robustness against deteriorated image quality.

## 4.3 Comparison on different dimensions

We next chose the top-4 models and show their quality on different dimensions, including domains, question types, image quality issues, and image resolutions.

---

[2]GPT-4.1 achieved an accuracy of 53%, though evaluation across other dimensions was not conducted.

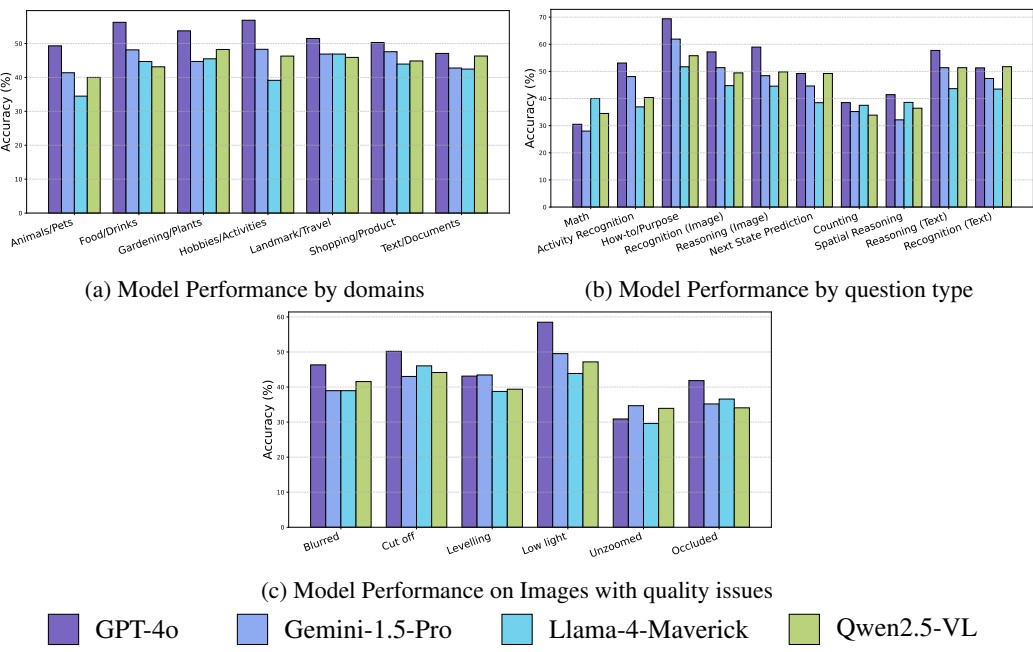

(a) Model Performance by domains

(b) Model Performance by question type

(c) Model Performance on Images with quality issues

■ GPT-4o    ■ Gemini-1.5-Pro    ■ Llama-4-Maverick    ■ Qwen2.5-VL

Figure 4: Performance of top-4 models across different dimensions, highlighting difficult user scenarios, and strengths and weaknesses of each model.

**Quality vs. domains:** Figure 4a shows performance of the models for different domains, and Table 6 in Appendix A.1 gives details for all models. We observe similar accuracy ranges across domains, whereas *Animals/pets* in general is the hardest domain, followed by *text* and *shopping*. The benchmark helps identify different strengths of different models: for example, GPT-4o dominates in domains like *animals, food, gardening,* and *hobbies*, but do not show major advantages in *landmark, shopping,* or *texts*; as another example, Llama-4-Maverick performs strong on *gardening* and *landmarks/travel*.

**Quality vs. question types:** Figure 4b shows performance of the models for different question types, and Table 7 in Appendix A.1 gives details for all models. The quality diversity is more pronounced along the dimension of question types. We observe much lower quality for *Object counting* (38% top-1), *Math* (40% top-1), and *Spatial Reasoning* (41% top-1), but much higher quality on *How-to questions* (69% top-1), *Image general reasoning* (59% top-1), *Image recognition* (58% top-1), and *Text Reasoning* (58% top-1). While GPT-4o is the top-1 for the majority of the question types, Llama-4-Maverick stands out in Math, and also performs well in *Object counting* and *Spatial reasoning*.

**Quality vs. images quality issues:** The benchmark incorporates 1,341 images with at least one quality issue, providing valuable insights into model robustness under challenging conditions. Figure 4c shows performance of the models with different quality issues, and Table 5 in Appendix A.1 gives details for all models. Among all quality issues, small images when *unzoomed* stands as the biggest challenge (35% top-1), since it is harder for recognition, especially for small texts. *Occlusion* presents another big challenge (42% top-1). Surprisingly, *Low-light* conditions exhibit minimal impact on QA quality, with models demonstrating superior performance (58% top-1); we hypothesis that this is because, in low light, the primary subject is more prominent with focused illumination, whereas other distracting elements are obscured by darkness.

**Quality vs. image resolution:** Finally, we focus on a subset of 413 images with high resolution (mostly 3024 x 4023), and experiment with two settings: low-resolution setting (960 x 1280) and mid-resolution setting (1536 x 2048). Table 4 compares the performance of top-4 models on the two resolution settings. Interestingly, we do not find clear patterns: 1) the performance drops on lower-resolution setting for Llama-4 and Gemini-1.5, but increases for Qwen2.5 and GPT-4o; 2) the

Table 4: QA accuracy (in percentages) of top-4 models on images of different resolutions. There is no clear pattern on how lower-resolution would impact VQA quality.

| Model | Overall | | Text/Documentations | | Visual Domains | |
|---|---|---|---|---|---|---|
| | Low-res | Mid-res | Low-res | Mid-res | Low-res | Mid-res |
| Llama-4-Maverick | 28.8 | 32.7 (+3.9) | 30.9 | 34.8 (+3.9) | 27.2 | 31.1 (+3.9) |
| Qwen2.5-VL | 38.0 | 36.1 ( -1.9) | 39.9 | 33.2 ( -6.7) | 36.6 | 38.3 (+1.7) |
| Gemini-1.5-Pro | 34.9 | 42.4 (+7.5) | 25.3 | 36.0 (+10.7) | 42.1 | 47.2 (+5.1) |
| GPT-4o | 43.8 | 41.7 ( -2.1) | 32.6 | 30.9 ( -1.5) | 52.3 | 49.8 ( -2.5) |

variations are bigger on text images than visual images for Qwen2.5 and Gemini-1.5, but similar for Llama-4 and GPT-4o; 3) on low-resolution images where text understanding intuitively is be more challenging, the performance drops significantly for Gemini-1.5 and GPT-4o compared to visual images, but increases slightly for Llama-4 and Qwen2.5. We suspect the performance highly rely on distribution of image resolutions in the training data used by different models.

## 4.4 Summary

To summarize, the evaluation along various dimensions reveals several performance gaps that warrant attention (**Q2**). 1) For object counting and mathematical reasoning tasks, there remains a need for more precise object localization and enumeration techniques. 2) Models also face challenges in spatial reasoning tasks, which demand a nuanced understanding of spatial relationships within images, highlighting the need for more sophisticated spatial awareness and reasoning capabilities. 3) For low-quality is images, particularly those that are unzoomed, or occluded, there is a marked performance decline compared to high-quality images. 4) Text and document understanding on low-resolution images can be challenging for certain models, and is worth tuning.

## 5 Limitations

One major limitation of the benchmark is the constraint of *no external-source needed*. A large portion of voice questions requires external information, and thus research on RAG (Retrieval-Augmented Generation) abounds. RAG for multi-model contexts (MM-RAG) has received significantly less attention [18], and needs a benchmark to guide its development. Another significant limitation of the benchmark lies in its exclusive focus on English-language queries, which potentially limits the benchmark's ability to evaluate models in diverse linguistic environments. Finally, the benchmark's reliance on static image-based design fails to adequately capture the temporal dynamics inherent in real-world scenarios, such as motion blur and fluctuating lighting conditions over time, as well as video reasoning capabilities.

## 6 Conclusion

The WearVQA Benchmark establishes a comprehensive and rigorous evaluation framework for multi-modal AI in the context of wearable technology, addressing significant gaps in existing benchmarks. By offering a diverse dataset of egocentric images and challenging question-answer pairs across various image classes, question types, and image-quality categories, the benchmark enables precise assessment of model robustness under real-world constraints. Our evaluation highlights the strengths of state-of-the-art multi-modal language models, such as GPT-4o, while also revealing persistent challenges in areas like spatial reasoning, counting, and handling occlusions. As we move forward, we plan to expand the benchmark to include multilingual support and dynamic video sequences, ensuring that WearVQA remains at the forefront of wearable AI research, adapts to emerging challenges, and evolves to meet new research needs.

## 7 Ethics Statement

Data collection and release were subject to Meta's internal privacy review and approval process.

**Bystander privacy in data collection** All images in the dataset were collected by paid participants who signed a data-collection agreement which specifies that the data may only be captured within approved locations. To protect privacy, all bystanders were irreversibly blurred using pixelation-based methods (not Gaussian filters, which can be reversible).

**Sensitive data removal** All images were reviewed to remove those containing potentially sensitive information, including but not limited to content related to child abuse, hate speech, dangerous organizations, mental health, or political affiliations.

**Demographic distribution of data collectors** Collectors were U.S. residents with diverse demographics as shown in Figure 5.

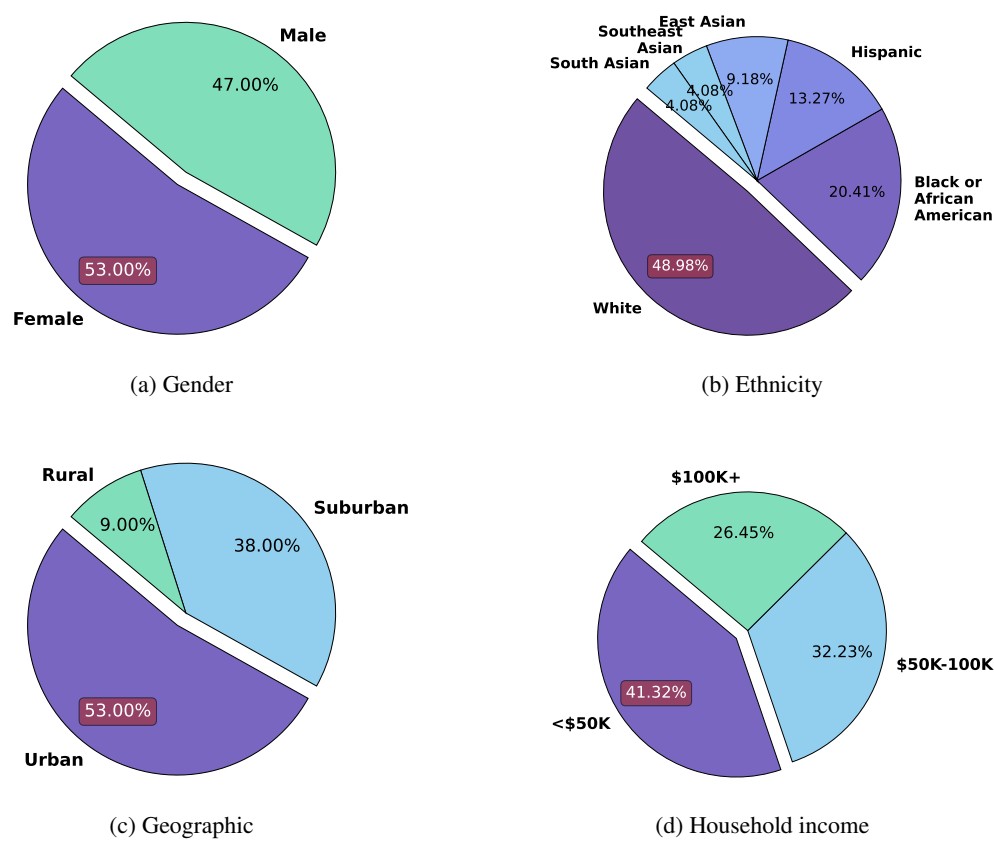

Figure 5: Demographic distribution of data collectors

**Limitations and intended use** This dataset is intended strictly for research purposes and benchmarking. It should not be used for model training or to infer personal attributes or identify individuals. We request that all derivative works properly cite this dataset and adhere to ethical standards regarding privacy and data protection.

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

# A  Appendix

## A.1  Model Performance on various dimensions

Table 5: Percentage of Errors on Images with Quality Issues across Models

| Model | At least one issue | Blurred | Cut off | Rotated | Low light | Unzoomed | Occluded |
|---|---|---|---|---|---|---|---|
| Claude-3.7-sonnet | 26.4 | 22.5 | 30.6 | 24.7 | 32.1 | 18.7 | 22.7 |
| Gemini-1.5-Pro | 41.0 | 39.0 | 43.0 | 43.4 | 49.5 | 34.7 | 35.2 |
| GPT-4o | 45.5 | 46.3 | 50.2 | 43.1 | 58.5 | 30.9 | 41.8 |
| Llama-3.2v | 33.8 | 32.0 | 37.4 | 34.1 | 42.9 | 25.3 | 30.5 |
| Llama-4-Maverick | 39.3 | 39.0 | 46.0 | 38.8 | 43.9 | 29.6 | 36.6 |
| Llama-4-Scout | 38.0 | 38.5 | 44.9 | 36.3 | 41.0 | 30.9 | 34.1 |
| Qwen2.5-VL | 39.9 | 41.6 | 44.2 | 39.4 | 47.2 | 33.9 | 34.1 |
| Molmo | 22.2 | 19.9 | 26.8 | 19.7 | 31.1 | 13.7 | 20.8 |
| Pixtral | 22.7 | 23.8 | 28.3 | 18.8 | 34.4 | 14.9 | 21.3 |
| Phi-4 | 21.3 | 19.1 | 24.5 | 21.3 | 28.3 | 13.7 | 19.9 |

Table 6: Performance comparison across different models and domains (in percentages).

| Model | Animals/ Pets | Food/ Drinks | Gardening/ Plants | Hobbies/ Activities | Landmark/ Travel | Shopping/ Product | Text/ Documents |
|---|---|---|---|---|---|---|---|
| Claude-3.7-sonnet | 30.0 | 36.3 | 39.2 | 36.3 | 36.4 | 33.9 | 30.0 |
| Gemini-1.5-Pro | 41.4 | 48.1 | 44.7 | 48.3 | 46.9 | 47.6 | 42.8 |
| GPT-4o | 49.3 | 56.3 | 53.7 | 56.9 | 51.5 | 50.3 | 47.1 |
| Llama-3.2v | 35.5 | 40.6 | 34.9 | 37.4 | 38.7 | 37.9 | 36.9 |
| Llama-4-Maverick | 34.5 | 44.7 | 45.5 | 39.1 | 46.9 | 43.9 | 42.5 |
| Llama-4-Scout | 31.7 | 47.8 | 40.8 | 39.7 | 45.3 | 44.6 | 40.8 |
| Qwen2.5-VL | 40.0 | 43.1 | 48.2 | 46.3 | 45.9 | 44.9 | 46.3 |
| Pixtral | 26.6 | 25.6 | 30.2 | 25.4 | 31.5 | 27.0 | 20.9 |
| Phi-4 | 20.0 | 29.4 | 27.5 | 21.7 | 25.6 | 25.2 | 21.4 |
| Molmo | 25.9 | 27.2 | 23.1 | 30.3 | 28.5 | 28.5 | 20.8 |

Table 7: Performance comparison across different question types and models (in percentages).

| Model | Activity Recognition | How-to/ Purpose | Recognition (Image) | Reasoning (Image) | Math | Next State Prediction | Counting | Spatial Reasoning | Reasoning (Text) | Recognition (Text) |
|---|---|---|---|---|---|---|---|---|---|---|
| Claude-3.7-sonnet | 35.8 | 56.5 | 34.8 | 40.7 | 23.5 | 23.1 | 17.1 | 30.4 | 34.1 | 31.7 |
| Gemini-1.5-Pro | 48.1 | 61.9 | 51.4 | 48.4 | 28.0 | 44.6 | 35.2 | 32.1 | 51.4 | 47.4 |
| GPT-4o | 53.1 | 69.4 | 57.2 | 59.0 | 30.5 | 49.2 | 38.5 | 41.4 | 57.7 | 51.3 |
| Llama-3.2v | 37.7 | 44.6 | 41.2 | 39.7 | 28.0 | 20.0 | 30.3 | 35.7 | 39.6 | 42.2 |
| Llama-4-Maverick | 36.9 | 51.7 | 44.8 | 44.6 | 40.0 | 38.5 | 37.5 | 38.6 | 43.6 | 43.5 |
| Llama-4-Scout | 36.5 | 53.4 | 42.8 | 42.1 | 36.0 | 41.5 | 36.5 | 38.6 | 44.6 | 41.3 |
| Qwen2.5-VL | 40.4 | 55.8 | 49.5 | 49.8 | 34.5 | 49.2 | 33.9 | 36.4 | 51.4 | 51.7 |
| Pixtral | 19.6 | 41.8 | 25.4 | 30.9 | 14.5 | 16.9 | 23.7 | 26.1 | 29.1 | 18.7 |
| Phi-4 | 18.5 | 25.9 | 30.4 | 22.8 | 9.5 | 16.9 | 31.3 | 19.3 | 25.9 | 27.4 |
| Molmo | 26.9 | 41.2 | 27.9 | 30.2 | 15.0 | 15.4 | 17.1 | 24.3 | 27.3 | 19.6 |

## A.2  Evaluation Prompt

The following prompt was used for all evaluations:

```
You are a multimodal assistant, designed to
    objectively evaluate the answer to
    image-related questions. Given the image and
    the question "{query}", please assess the
    response: "{response}" against the ground
    truth: "{gt}". Your evaluation should be based
    on the following criteria:
- Relevance: Does the response make sense in the
    context of the image and the question? If the
    response is incoherent, irrelevant, or fails to
    answer the question (e.g., "I don't know", "I
```

```
          can't help you with that"), assign a grade of
          false.
      - Correctness: Is the response accurate compared
          to both the image and the ground truth? Note
          that the response doesn't need to contain all
          the information in the ground truth, but it
          must correctly answer the question. For textual
          information in the image, only evaluate if the
          response recognizes the text correctly, without
          judging its validity.
      - Ego-centric: The response should be phrased in
          an ego-centric way. This means that the model
          should not regard the image as an image but as
          a person's point of view. For example, if the
          image shows a cat, the correct response would
          be "There is a cat" instead of "The image has a
          cat". If the format is incorrect, assign a
          grade of false.
      - Conciseness: Does the response contain anything
          not directly related to the question? It is
          okay for the response to elaborate in detail as
          long as the response is answering the question.
          But if the response contains irrelevant
          information such as greetings and side-tracks,
          assign a grade of false.
      Evaluation Guidelines:
      - Assign a grade of true if the response meets all
          the above criteria.
      - Assign a grade of false if any of the above
          criteria are not met.
      - Provide a detailed explanation for your grade,
          going over the above criteria one-by-one.
      Required Response Format:
      Please respond with a JSON object in the following
          format: {"grade": [true or false], "reason":
          [brief explanation]}
      Note: Only include the required JSON object in
          your response.
```

### A.3 Human Data Collection Protocol

**Compensation** All participants were paid a minimum of 35 USD per hour, exceeding minimum wage.

**Instructions** The following instructions were shared with data collectors (condensed for brevity).

Overall participant demographics: Please do not recruit participants from Washington, Illinois or Texas. We cannot use data recorded in those states or from residents of those states. Multimodal AI is enabled for all other areas of the United States and Canada.

Quality Specifications: To meet the minimum volume for this project, we require that the captures you share with us meet our minimum quality specifications. Captures you share with us that do not meet our minimum quality specifications will not count toward your total capture volume. To ensure you are able to meet these requirements, we encourage you to over capture – we will review your collected content and provide feedback to help you understand your performance. Before your recording session, please capture a short (10-30 second) video and sync it back to your device to make sure your lens is not occluded. We will provide continuous quality feedback for your recordings. This will be communicated to you by your manager. Please do your best to record quality videos for this collection.

Topics Not to Record: Some topics should never be recorded for this collection. Please use good judgment when taking captures and immediately delete any violating content that you might accidentally record in this process. Violating content includes anything illegal, any personally identifying information, such as contact information, financial information, medical or legal information, or any sexual, illegal, or violent content. Please use the table below as a guide, but keep in mind that the list of violating content in this table is not exhaustive. Use your best judgment, and do not record anything that you think might count as violating.

Places not to record: There are some locations where you should never record content because of the high likelihood of recording violating content in those locations. The provided devices record audio/image/video content, as well as other types of data (including location data and device usage data), as further described in the Research Participation and Data Collection Agreement. You will turn off the device recording functionality in the following situations: (1) when involved in or observing a private conversation or situation, whether in person, via phone, or otherwise, where the parties have a reasonable expectation that they will not be overheard or recorded; (2) in intimate or sensitive spaces (such as restrooms and locker rooms, prayer rooms/active houses of worship, mother's rooms, government facilities, medical facilities, courthouses, schools/daycares, adult entertainment facilities, political conventions, rallies, or protests); (3) in the presence of minors; (4) when requested by a bystander or recording is otherwise prohibited; (5) in any other situation in which individuals may have a reasonable expectation of privacy (such as when discussing sensitive personal information like passwords, banking details, social security numbers, or PSC ratings); and (6) in accordance with any training you may receive as a Participant in the Research.

If you mistakenly do not turn off the recording functionality in a situation described above, you will immediately delete any such recorded information. If you mistakenly do not delete such recorded information, or encounter any issues regarding the above instruction, you will immediately notify your Research Administrator so that such information can be deleted.

