# OpenReview forum: "WearVQA: A Visual Question Answering Benchmark for Wearables in Egocentric Authentic Real-world scenarios"
_NeurIPS.cc/2025/Datasets_and_Benchmarks_Track — NeurIPS 2025 Datasets and Benchmarks Track poster_

### Official Review · Reviewer_haD9 · 2025-07-01

**Rating:** 4
**Confidence:** 4

**Summary:**

The paper contributes:

- a diverse egocentric VQA dataset towards enabling benchmarking of multi-modal LLMs on AI-powered glass-based wearables.
    - this includes: 2.5k image-QA triplets across 7 different domains (shopping, travel, gardening, animals, etc.) across 10 types of questions – with 5 types of image quality degradations (occlusions, rotations, blurriness, etc.) to test robustness
- an LLM-based evaluation scheme to automate the evaluation of answers of the variety of multi-modal LLMs they are benchmarking
- benchmarking a suite of open-source and proprietary LLMs on this dataset – showing how current models are lagging in object localization, enumeration, spatial awareness, and reasoning capabilities.

**Additional Feedback:**

- L36: It is not clear to me how the contributed dataset enable "sequential or causal reasoning" with single-frame-based questions.
- I wouldn't call features at the end of page 2 "distinctive".

**Dataset Code Accessibility:**

No

**Dataset Code Comments:**

Couldn't find a link to the dataset anywhere in the paper. Also did a google search for the dataset's name and didn't find anything.

**Ethical Considerations:**

No, there are no or only very minor ethics concerns

**Final Justification:**

Barring my gripe with the benchmark not including multiple frames (even though they were already captured during data collection), I think this paper is a valuable addition to the community and will possibly be a good first-step towards better VLM video understanding abilities.

**Limitations Weaknesses:**

- Lack of temporal support (as also acknowledged by the authors in the limitations section) seems to be a major concern for me. The temporal dimension should theoretically only improve performance of models (if utilized in the right way) – especially for the low quality images (assuming that preceding and succeeding frames often could be blur-free, won't get cut off, etc.). It does not seem realistic that any models deployed on these wearables will not have access to a sequence of frames (before and after the current frame). Therefore, probing these models with a sequence might better reflect the current state of these models.
- Also, because these images were extracted from videos captured by annotators, I don't see a reason to not include n frames before and after the current frame – for each question-answer pair in the dataset. With that kind of a dataset, one could have a single-image based evaluation as an additional experiment. If the goal is to evaluate the performance of these models in wearable-settings, why not leverage useful (and readily available) frame sequence context?
---
- Along similar lines, the omission of questions from external sources also makes the coverage across questions from real-world (wearable) applications smaller, but that could understably be less tractable and out of scope of this paper.
- Apart from that, I don't know how I feel about removing examples from the dataset (that fall in the natural distribution of questions these models will be prompted for in wearable applications) that a lot of these models are good at – to engineer the benchmark to be more challenging.
- Also, it wasn't clear to me why next-state prediction questions are fewer and claimed to be "unusual in real scenarios" (L183-184). E.g. "what to do with XYZ?" or "how to use XYZ?" seem to be just as common (if not more) as "what is the word above XYZ?"
- It is also not clear to me why the EgoVQA and VizWiz datasets were not included in Table 1 – given that they were probably the most similar to the WearVQA dataset (and both egocentric; as opposed to all non-egocentric datasets listed in Table 1) and why there wasn't adequate discussion about their differences against WearVQA.

**Strengths Contributions:**

I think the dataset is a valuable test-bed to evaluate single-image-based reasoning capabilities of current LLMs. This can be a good starting point to test the abilities of these large models on individual frames (with natural noise and degradation) – before eventually giving them more context with neighboring frames.

- The paper reads easy, is well organized, answers most questions readers might have about dataset collection, annotation, and its distribution statistics. It also seems to be comprehensive in comparing existing datasets and the suite of LLMs.
- Leaving aside the fact that questions from external sources are not included, I think the set of image-QA triples are pretty diverse in the types of questions, the domains they cover and the types of image degradation issues they include from real-world usage.
- The breakdown of model performance across domains, question types, quality issues, and image resolutions seems pretty useful in identifying areas of improvement along each dimension of reasoning.

---

> ### Author Rebuttal · Authors · 2025-07-30
>
> We thank the reviewers for recognizing the potential impact of WearVQA in advancing wearable AI experiences through a dedicated wearables-representative benchmark. We appreciate the constructive feedback and, below, address the raised concerns and offer clarifications.
>
> **W1, W2:** Lack of temporal support using before/after frames. Expect wearable devices to have access to those.
>
> **A1/2:** Our benchmark uses single images in order to mimic real wearables usage. Wearable devices (e.g., RayBan Meta) oftentimes do not capture or store temporal information when responding to a visual query, mainly due to power consumption as well as bandwidth and latency limitations of transmitting images to the server. When a user asks a question, the device takes a single snapshot of the visual field, and no before/after frames are captured.
>
> According to Shenoy et al., (2024): just transferring a high-resolution image from device to cloud costs significant time resulting in a poor user experience. For instance, transmitting an image of size 3𝑘 × 4𝑘 (standard resolution for today’s devices) from a device to the cloud may take several seconds before even running any AI models. And the end-to-end time to get a response would be even longer, making for a poor experience.
>
> In the future, we are planning to build a video QA benchmark that focuses on video reasoning capabilities for live sessions. This will be considerably more challenging in designing the questions and ensuring QA accuracy. We will leave this for future work and will include it in the Limitation section.
>
> **W3:** Dataset does not include questions that require external knowledge.
>
> We agree that not including RAG questions is indeed a limitation, as we call out in the Limitation section. Designing a multimodal RAG benchmark involves tremendous efforts and cannot be easily incorporated into this paper. Given that there are several VQA benchmarks that only require common sense (e.g., VQA V2, Visual7W, GQA), we believe that this paper has sufficient contributions and merit of its own.
>
> We would also like to mention that the team that developed WearVQA has built another wearable VQA benchmark called CRAG-MM (part of KDD Cup 2025), which is designed to test RAG capabilities. While the two benchmarks are currently kept separate to highlight the different challenges , we will release both to help the research community develop robust VQA solutions with full coverage of visual questions.
>
> **W4:** Reviewer objects to removing easy examples to engineer the benchmark to be more challenging.
>
> **A4:** We only remove a question if at least 5 (out of 10 models we have evaluated in Section 4) models answer it correctly, demonstrating that the question adds little value to the benchmark and it is difficult to demonstrate hill climb on it. While the easy, recognition-based questions may be more representative of overall wearables use cases, they are not representative of the majority of the failure cases. We will consider releasing the removed questions as a separate set.
>
> **W5:** Reviewer objects to undersampling “next-state prediction” questions and mentions that “how to use XYZ” types of questions are presumably common in wearables use cases.
>
> **A5:** In our benchmark, “what to do with XYZ” and “how to use XYZ” belong in the “how-to/purpose” question type (not “next-state prediction”), and those are sampled with a ratio on-par with other question types (~12% of the questions in the benchmark are how-to/purpose).
>
> Examples of “next-state prediction” questions include:
> - (For an image of Lego): Which step in the manual should I follow next?
> - (For an image of Jenga): What will happen if I pull out the block in the middle on the fourth level from the table?
>
> While these questions do occur in real life usage, we have found those to be  relatively rare as most users focus on understanding and reasoning over their current field of view.
>
> **W6:** Include EgoVQA and VizWiz in Table 1 and discuss the differences with WearVQA.
>
> **A6:** We will include VizWiz in Table 1. EgoVQA, however, is a video-based QA and fundamentally different from our work. We will mention such video-based QA benchmarks in the revision as well.
>
>  Aside from that, the key differentiators for WearVQA against these two benchmarks are:
> - Low-quality images: While EgoVQA mentions video blur as a potential image quality issue, it does not have a dedicated subset that incorporates other common wearables image quality issues such as occlusion, tilt, etc.
> - Question diversity: The questions from EgoVQA primarily consist of activity recognition (what am I doing) and object recognition (what is that person holding), whereas WearVQA includes more diverse question types including how-to, math, and text reasoning.
> - Reasoning complexity: While VizWiz also provides low quality images and diverse images/questions, most questions are recognition-related (e.g. What color is this? What is this item?), as opposed to requiring complex reasoning, due to the purpose of the collection (answering visual questions from people who are blind).

---

> > ### Author Response · Authors · 2025-08-07
> >
> > Thank you again for your feedback. We would appreciate any additional feedback or questions in light of our response.

---

### Official Review · Reviewer_1eC2 · 2025-07-01

**Rating:** 5
**Confidence:** 4

**Summary:**

WearVQA proposed the first VQA dataset for wearable devices, which uses egocentric interaction and includes image problems such as image blur that may be encountered during wearable device use. WearVQA covers multiple types of images and cognitive task types.

**Additional Feedback:**

See Limitations.

**Dataset Code Accessibility:**

NA; not applicable to this submission (e.g., no new dataset, benchmark, code, or data provided)

**Ethical Comments:**

WearVQA used RayBan Meta smart glasses to capture ego centric images, and the process of annotators using smart glasses for collection may involve the privacy of others. The author needs to explain whether there are any issues related to the privacy and authorization of others, especially in Activity Recognition.

**Ethical Considerations:**

Yes, there are significant ethics concerns that require review by an ethics expert

**Ethics Flags:**

["Data privacy, copyright, and consent", "Human rights (including surveillance)"]

**Final Justification:**

The author's response addressed my concerns. WearVQA is a very meaningful dataset. I hope the author will address the aforementioned issues in the revised version.

**Limitations Weaknesses:**

1. The main drawback of WearVQA is that all of its questions can only be answered using images and common sense, without the need for external information, which is somewhat disconnected from real-life voice problems. The author can develop an extended version to add some questions that require external knowledge to answer in the dataset.
2. Another drawback is that all questions are set to standard VQA settings. In real-world speech problems, there is a fuzzy matching problem with fuzzy inputs, and the author can perform fuzzy processing on the questions in the dataset.
3. Standardize and unify commonly used terms in papers, such as "ego centric" vs "egocentric", "unimagified" vs "unzoomed".

**Strengths Contributions:**

1. The key advantage of this work lies in its wearable devices - an area that is rapidly developing and requires benchmark guidance.
2. The author proposed a benchmark based on wearable devices, which is a valuable contribution.
3. Benchmarks cover multiple fields and problems, and have been thoroughly tested using state-of-the-art multimodal language models. The current model has encountered many challenges, especially in inference, comprehension, and image quality issues.
4. The previous dataset did not possess features such as Egocentric view and Low Quality Images, which are crucial for the development of wearable device VQA.
5. The paper is well written and easy to understand.

---

> ### Author Rebuttal · Authors · 2025-07-30
>
> We thank the reviewers for recognizing the potential impact of WearVQA in advancing wearable AI experiences through a dedicated wearables-representative benchmark. We appreciate the constructive feedback and, below, address the raised concerns and offer clarifications.
>
> **W1:** Dataset does not include questions that require external information.
>
> **A1:** We agree that not including RAG questions is indeed a limitation, as we call out in the Limitation section. Designing a multimodal RAG benchmark involves tremendous efforts and cannot be easily incorporated into this paper. Given that there are several VQA benchmarks that only require common sense (e.g., VQA V2, Visual7W, GQA), we believe that this paper has sufficient contributions and merit of its own.
>
> We would also like to mention that the team that developed WearVQA has built another wearable VQA benchmark called CRAG-MM (part of KDD Cup 2025), which is designed to test RAG capabilities. While the two benchmarks are currently kept separate to highlight the different challenges, we will release both to help the research community develop robust VQA solutions with full coverage of visual questions.
>
> **W2:** Benchmark should take into account fuzziness in speech/ASR.
>
> **A2:** We assume the reviewer means fuzziness from voice understanding. We also believe that speech/ASR quality is one of the unique challenges of wearable devices, similar to image quality. We opted to leave question fuzziness out of the scope of this benchmark in order to focus on VQA, as question fuzziness can dilute this theme. However, we will include this in the Limitation section and will consider extensions in future versions.
>
> If the reviewer has in mind other types of fuzziness, we are happy to provide answers.
>
> **W3:** Standardize common terms, e.g. “ego centric” vs. “egocentric.
>
> **A3:** We will standardize the terminology throughout the paper. Thank you very much for pointing it out.

---

> > ### Author Response · Authors · 2025-08-07
> >
> > Thank you again for your feedback. We would appreciate any additional feedback or questions in light of our response.

---

> > ### Comment · Reviewer_1eC2 · 2025-08-07
> > **Response to Rebuttal**
> >
> > The author's response addressed my concerns. WearVQA is a very meaningful dataset. I hope the author will address the aforementioned issues in the revised version.

---

### Official Review · Reviewer_eCf1 · 2025-07-03

**Rating:** 5
**Confidence:** 4

**Summary:**

In this paper, the authors propose WearVQA, a benchmark targeted to evaluate the visual question answering (VQA) capabilities of existing VLMs with egocetric images as inputs. The benchmark comprises 2,520 image-question-answer triplets and exsiting VLMs can only achieve a accurate as low as 24–52%.

**Dataset Code Accessibility:**

No

**Dataset Code Comments:**

I have not seen the link for the code or data.

**Ethical Considerations:**

No, there are no or only very minor ethics concerns

**Final Justification:**

After reading the authors' rebuttal and other reviews, I have no concerns for the paper and recommend the accecption for the paper.

**Limitations Weaknesses:**

1. Compared to other benchmarks like MMQA and MMMU (more than 10000 images), the number of the images in this benchmark is small (2520). Since the benchmark covers many aspects, it is expected to increase the dataset size.
2. For the VQA task, it seems there is no essential difference beteew egocentric image and other image. Can authors present more examples of  typical egocentric images?

**Strengths Contributions:**

1. The authors propose the first benchmark to evaluate VQA abilities for VLMs in the aspect of egocetric recognition. And the benchmark results show that  existing  VLM can only achieve accuracy between 24%-52%, which illustates the necessity of the benchmark.
2. The questions of the benchmark are well-curated,  covering 10 visual cognitive task types. And the authors also consider the egocentric image quality variations, simulating the first-person perspective.
3. The experiments cover a variaty of aspects to show the abilities of current VLMs. These results can inspire future research to improve the existing VLMs.

---

> ### Author Rebuttal · Authors · 2025-07-30
>
> We thank the reviewers for recognizing the potential impact of WearVQA in advancing wearable AI experiences through a dedicated wearables-representative benchmark. We appreciate the constructive feedback and, below, address the raised concerns and offer clarifications.
>
> **W1:** Dataset size is relatively small.
>
> **A1:** Even with the current volume, the margin of error is capped at 5.7% at 95% confidence level for all image quality buckets and question types (except “next-state prediction” which has 12% MoE, but we expect this question type to be uncommon in real scenarios as stated in the paper).
>
> **W2:** It seems there is no essential difference between egocentric and other images.
>
>
> **A2:** The core defining aspects of egocentric images include
> - Images are from the user’s perspective, oftentimes with wide angle and small main entities; See the “Unmagnified” image in Figure 2 as an example.
> - Poor image quality (blur, occlusion, composition, etc.); see Figure 2 for examples.
> - Lower resolution.
>
> All of the above aspects are represented in our benchmark. In particular, we have ensured that 54% of the images in the benchmark contain at least one of the common wearables image quality issues that extend beyond RayBan Meta. 83% of the images are mid-resolution (720 x 1280) which is the typical size of images that can be easily transferred to the server through wifi.
>
> We will highlight such distinctions in the next version of our paper.

---

> > ### Author Response · Authors · 2025-08-07
> >
> > Thank you again for your feedback. We would appreciate any additional feedback or questions in light of our response.

---

### Official Review · Reviewer_pDVf · 2025-07-03

**Rating:** 4
**Confidence:** 4

**Summary:**

This paper introduces the first benchmark for evaluating the visual question answering ability of multimodal AI experts in self-centered wearable scenarios in the real world, wearVQA. This benchmark is comprehensive and strict. This benchmark designed 10 types of questions to evaluate the model's understanding and reasoning abilities. Images and questions are obtained or constructed according to a self-centered requirement. This benchmark effectively reveals the advantages and challenges of state-of-the-art multimodal large models and is expected to provide development directions for future research.

**Dataset Code Accessibility:**

No

**Dataset Code Comments:**

The dataset and source code do not appear to be publicly available.

**Ethical Considerations:**

No, there are no or only very minor ethics concerns

**Final Justification:**

The authors have addressed my concerns, and I have no more questions.

**Limitations Weaknesses:**

1. Has this work considered the issue of language bias in VQA? In section 2.3, this work considered domain distribution, problem type distribution, and image quality distribution, but did not take into account the distribution of answers. Due to the poor quality of some images, the model may answer frequently based on the question. Providing answer distribution (including dataset level and question type level) can address this concern.
2. As wearable devices are diverse, will different wearable devices have a significant impact? The author can conduct experiments on more wearable devices.

**Strengths Contributions:**

1. This work is interesting, which aims to address the visual question answering ability of multimodal large models in wearable scenarios, and carefully designs many constraints for comprehensive evaluation. The experimental results demonstrate that the work is meaningful and provides valuable resources for further research on large models in real-world application scenarios.
2. This work had a clear discussion with previous works such as VQA v2, OK-VQA, MMQA, etc., pointing out the shortcomings of the previous work and highlighting the importance and value of this work.
3. This paper is well-organized, written clearly, and easy to understand.

---

> ### Author Rebuttal · Authors · 2025-07-30
>
> We thank the reviewers for recognizing the potential impact of WearVQA in advancing wearable AI experiences through a dedicated wearables-representative benchmark. We appreciate the constructive feedback and, below, address the raised concerns and offer clarifications.
>
> **W1:** Provide answer distribution (at dataset level and question type level) to show the answers are not biased by questions.
>
> **A1:** We have designed the QA pairs to be freeform, as opposed to categorical. For example:
> - Q: “What are the two meat products on the table?” / A: “The two meat products on the table are pepperoni and smoked turkey.”
> - Q: “What is the solution to this math problem?” / A: “The solution is 120.”
> - Q: “In which state is the valley named on the record player located?” / A: “Coachella Valley is located in California.”
>
> Given that each answer is unique and freeform, it is not straightforward to calculate a distribution. If the reviewer is more specific on what kind of distribution they have in mind, we would be happy to provide it.
>
> **W2:** Will different wearable devices have a significant impact?
>
> **A2:** We expect the images in our benchmark to be representative of wearable devices in general. The core defining aspects of wearable media are (1) egocentricity (2) poor image quality (blur, occlusion, composition, etc.), and (3) lower resolution, all of which are explicitly represented in our benchmark. In particular, we have ensured that 54% of the images in the benchmark contain at least one of the common wearables image quality issues that extend beyond RayBan Meta. 83% of the images are mid-resolution (720 x 1280) which is the typical size of images that can be easily transferred to the server through wifi.

---

> > ### Author Response · Authors · 2025-08-07
> >
> > Thank you again for your feedback. We would appreciate any additional feedback or questions in light of our response.

---

### Note · Authors · 2025-08-12

We thank the reviewers for recognizing the potential impact of WearVQA in advancing wearable AI and for their insightful feedback. Below we provide final pressing comments.


**Ethics**

[Mj5C, DKBH, 1eC2] We followed Meta’s rigorous privacy approval process for data release. Measures included:
- Bystander privacy: Collectors signed agreements prohibiting recording bystanders unless they also signed; all faces blurred with irreversible methods.
- Sensitive data: Every image reviewed to ensure no harmful or sensitive content.
- Collector demographics: All based in the U.S., with diverse demographics (gender, ethnicity, geography, income)


**Scope**

We made careful decisions to leave the following aspects for future work.
- [haD9-W3, 1eC2-W1] **RAG questions:** Designing a MM-RAG benchmark involves tremendous efforts and is hard to incorporate into this paper. As there are VQA benchmarks that require only common knowledge (e.g., VQA V2, Visual7W, GQA), we believe this paper has sufficient merit of its own. The team also developed CRAG-MM (KDD Cup 2025) for RAG evaluation; both datasets will be released to cover complementary challenges.
- [haD930] **Temporal support:** WearVQA contains only a single image for each question to mimic real wearables use cases. Wearables devices (e.g., Ray-Ban Meta) typically do not store temporal sequences because of power, bandwidth, and latency constraints. When a query is asked, a single snapshot is taken and transferred to the cloud for QA.


**Representativeness**

We believe our benchmark is representative of challenges faced by wearables use cases and allows reliable evaluation.
- [pDVf03] Though we used Ray-Ban Meta to construct WearVQA, it reflects core wearable media traits: (1) egocentric view, (2) low image quality (54% images), and (3) lower resolution (83% images).
- [haD930] We removed examples where >=5 models answered correctly, as we consider them not representing key challenges faced by wearables. - We will consider releasing the removed questions as a separate set.
- [eCf103] Despite its seemingly small size (2,500 questions), MoE is capped at 5.7% w. 95% conf. for all question type and image quality buckets.


**Comparisons with EgoVQA/VizWiz**

[haD930] WearVQA is fundamentally different from VizWiz and EgoVQA (video-based QA) by:
- Low-quality images (occlusion, tilt, etc.)
- Question diversity (how-to, math, and text reasoning etc.)
- Reasoning complexity (60% questions)

---

### Decision · Program_Chairs · 2025-09-18

**Decision:**

Accept (poster)

**Comment:**

Reviewers rating includes two Accept and two Borderline accept. After reviewing the discussions I tend to agree with the reviewers and would like to see this paper to be Accepted. Here is a summary of contributions:

The WearVQA benchmark was developed under Meta’s strict privacy guidelines, ensuring bystander protection, sensitive content screening, and diverse U.S.-based data collection. It focuses on real wearable device scenarios, using single low-quality, egocentric images without temporal sequences due to practical constraints, and excludes overly easy questions to better capture real-world challenges. Although limited to 2,500 questions, it maintains statistical reliability and differs from datasets like VizWiz and EgoVQA through its emphasis on challenging, diverse question types and complex reasoning. Some aspects, such as multimodal RAG benchmarks and temporal support, are left for future work, with complementary datasets like CRAG-MM planned for release.